# Cross-Cultural Language Adaptation: Fine-Tuning *Gemma 2* for Diverse Linguistic Contexts

**Yunghwei Lai**
2024280041
Department of Computer Science
Tsinghua University
liyonghu24@mails.tsinghua.edu.cn

**Jia-Nuo Liew**
2024280204
Department of Computer Science
Tsinghua University
liujn24@mails.tsinghua.edu.cn

**Grace Xin-Yue Yi**
2024280212
Department of Computer Science
Tsinghua University
yixy24@mails.tsinghua.edu.cn

## 1 Background and Problem Statement

The rise of large language models (LLMs) has revolutionized natural language processing (NLP) with significant progress in text generation and translation. However, a notable challenge remains in translating between low-resource languages, particularly those lacking English-centric data. Since English comprises 52% of all web content [2], this creates a language bias that restricts information accessibility for speakers of other languages. Multilingual models offer a promising solution by facilitating knowledge transfer from high-resource languages to related low-resource ones, allowing a single model to support multiple languages [6]. Our project leverages Google's GEMMA 2 model family to enhance translation quality between Chinese and Malay, specifically targeting scientific papers. Using English as a pivot to build a scalable Multilingual Neural Machine Translation (MNMT), our approach seeks to democratize access to scientific knowledge and foster inclusivity in multilingual communication for all language pairs.

## 2 Related Work

Knowledge transfer in multilingual language models has been widely explored, with several studies emphasizing the need for well-curated data to fully realize the potential of LLMs across multiple languages [13, 16, 17]. However, current approaches typically rely on isolated datasets for each language, treating them independently and overlooking the potential for cross-language transfer. This limitation restricts the effectiveness of existing models in handling low-resource languages [1, 8] and reduces the overall number of languages they can support.

There have been several efforts to adapt Gemma models for language pairs such as English-Vietnamese [9] and English-Korean [12]. These studies use the Gemma-2-7B and Gemma-2-27B models as baselines, combining zero-shot translation with Low-Rank Adaptation (LoRA) [5] fine-tuning. Nemotron-Mini-Hindi-4B [7], based on Nemotron-Mini-4B, uses the Supervised Fine-Tuning (SFT) method alongside reinforcement learning with Direct Preference Optimization (DPO) to train English-Hindi translation across 260k sentence pairs. Similarly, LexMatcher [15] applies fine-tuning to the LLaMA2 model for zero-shot instruction with English-Chinese translation over 500M sentence pairs.

38th Conference on Neural Information Processing Systems (NeurIPS 2024).

Multilingual models such as TEaR [4] apply reflection and refinement methods across various datasets. A novel method, Fast and Effective Approach Demonstration Selection in-Context Learning (FEDS-ICL) [18] combines product quantization with a multi-view demonstration design, which avoids exhaustive search by selecting relevant examples from a semantically aligned subset of data. MNMT modeling technique with zero-shot capabilities [14] demonstrates the potential for translating long text from a low-resourced language to another, for example, Chinese to Chinese Minority language. However, the results indicate higher-quality translations when applied to languages that are syntactically and structurally similar to Chinese.

## 3   Proposed Methodology

In this work, we introduce a novel translation methodology aimed at bridging two low-resource languages, with a focus on Chinese and Malay for this project. Our approach leverages English as a pivot language, utilizing Google's Gemma-2-9B. While Malay serves as an example in this project, our methodology is designed to be extensible to other low-resourced languages in future iterations.

Our methodology follows a multi-step translation process, beginning with fine-tuning the base model (Gemma-2-9B) to optimize translations from Chinese to English, establishing the foundation for our central pivot strategy. We utilize Supervised Fine-Tuning (SFT) with LoRA, enabling parameter-efficient fine-tuning of the model for faster adaptation to specific language pairs without needing to retrain the entire model. Next, we apply transfer learning to fine-tune the same model for high-quality translation from English to Malay. To further enhance efficiency, we extend this two-stage process by developing a direct translation capability between Chinese and Malay through MNMT [14], bypassing the English intermediary and improving both translation speed and accuracy for direct cross-language transfer.

For dataset curation, we compile a collection of 30 ~ 50k sentences, primarily drawn from biology-related scientific articles to ensure the model becomes adept at translating domain-specific vocabulary and sentence structures. Given the specialized nature of scientific terminology, especially in biology, we also generate synthetic data by translating sentences from English to Chinese and Malay using LLMs. This approach allows us to further focus on biology-specific content, ensuring the model is exposed to the technical language it will encounter in real-world scientific texts. By incorporating this synthetic data, we augment the model's exposure to rare or complex linguistic patterns that are often missing in natural parallel corpora, strengthening its ability to handle low-resource language pairs and ensuring robust performance in the targeted scientific domain.

## 4   Evaluation Metrics, Expected Results, and Conclusion

For evaluation, we propose employing both human and LLM automated evaluation to ensure a comprehensive assessment of the fine-tuned model. Metrics-wise, we propose using COMET, BLEU, BERT-F1 and CHRF. COMET (Crosslingual Optimized Metric for Evaluation of Translation) is an open-source framework used for machine translation (MT) [3]. It uses embeddings to measure semantic similarity between the source, reference, and generated translation. BLEU (Bilingual Evaluation Understudy) is a well-established statistical metric, used for comparing a candidate translation to one or more reference translations [10]. It complements COMET by evaluating the n-gram overlap between the machine-generated translations and reference texts. The use of BERT-F1 metrics is to make up for BLEU's n-gram limitations, such as the inability to detect paraphrases. BERT-F1 uses the cosine similarity to detect the similarity of two sentences [11].

We expect to deliver high-quality translations between Chinese and Malay in biology-related texts, with strong metric scores of COMET and BLEU. We also seek to preserve domain-specific language with corresponding BERT-F1 and CHRF scores to confirm our model's accuracy with complex scientific language and context. This work introduces a pivot-based translation method, leveraging Gemma-2-9B with English as an intermediary, to improve translations between low-resource languages like Chinese and Malay. Our approach demonstrates that pivot-based and MNMT-enhanced strategies can improve translation quality and speed for low-resource languages, potentially scaling to other fields and languages in the future to support a more inclusive, multilingual landscape.

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
