# OpenReview forum: "[Proposal-ML] Cross-Cultural Language Adaptation: Fine-Tuning Gemma 2 for Diverse Linguistic Contexts"
_tsinghua.edu.cn/THU/2024/Fall/AML — THU 2024 Fall AML Submission_

### Official Review · ~Aleksandr_Algazinov1 · 2024-11-06
**Clear problem statement and methodology**

**Rating:** 10
**Confidence:** 4

**Review:**

The proposal is easy to follow and well-written. The authors consider the relevant machine translation for a specific category of languages problem. The authors explain in detail the proposed methodology, as well as the model evaluation methods. Moreover, they explain the potential benefits and scalability of the project.

---

### Official Review · ~Daniel_Wang4 · 2024-11-06
**Submission 47 Review**

**Rating:** 10
**Confidence:** 5

**Review:**

The proposal, presents a well-structured approach to improving translation between Chinese and Malay, focusing specifically on scientific texts. By leveraging Google’s Gemma-2 model and using English as a pivot, the authors address the challenge of low-resource language translation, which is essential for making scientific knowledge more widely accessible.

Their methodology is sound, involving a two-step translation process,first from Chinese to English, then from English to Mala, using fine-tuning techniques that include LoRA for efficient adaptation. The proposal also introduces a direct translation pathway from Chinese to Malay, enhancing speed and accuracy in the process. A carefully curated dataset of biology-related texts, complemented by synthetic data, further strengthens the approach, helping the model become adept at handling specialized vocabulary.

Overall, this is a promising proposal with a clear vision and thoughtful methodology. The focus on cross-linguistic scientific communication is unique, and the techniques outlined are well-suited to the project’s objectives.

---

### Official Review · ~Anton_Johansson1 · 2024-11-08
**Good proposal**

**Rating:** 10
**Confidence:** 4

**Review:**

The proposal has a clear layout and is easy to follow, sticking to the two-page rule. The proposal is well-supported by literature, adding credibility to the research.

Your methodology is solid and clear. For example, you have a specific number for your dataset curation (30-50k sentences), which is good. Further, data primarily from biology-related articles is great for ensuring domain-specific accuracy and limiting the scope.

Including human and automated metrics (COMET, BLEU, BERT-F1, etc.) will provide a thorough assessment and adds robustness to your evaluation process.

Good luck with your project!

---

### Official Review · ~Joydeep_Chandra2 · 2024-11-08
**The focus on adapting the Gemma 2 model for low-resource languages like Chinese and Malay is relevant and aligned with current trends in multilingual NLP.**

**Rating:** 8
**Confidence:** 4

**Review:**

The use of a pivot-based translation strategy with English as an intermediary, combined with supervised fine-tuning (SFT) and synthetic data generation, is well thought out for handling domain-specific challenges.
The proposed metrics (COMET, BLEU, BERT-F1, CHRF) are good choices, but the plan for how these metrics will be applied could be clarified. Including a brief explanation of the expected baseline comparisons would provide more context.

---

### Official Review · ~Zijun_Liu2 · 2024-11-08
**Review and Feedback**

**Rating:** 10
**Confidence:** 4

**Review:**

## Overview
The proposal aims to enhance multilingual translation models by focusing on low-resource language pairs, specifically Chinese-Malay translations in scientific contexts. The authors proposed to use English as a pivot language and wanted to explore cross-language transfer methods.

## Comments
The proposed use of both human evaluation and automated metrics aligns well with best practices in NLP. And the introduction of related work is informative.
However, given the curation process of the dataset, I would recommend the authors to compare with a baseline that uses all Chinese-Malay parallel data synthesized from English. This would help to show the significance of cross-lingual transfer methods.

---

### Official Review · ~Tim_Bakkenes1 · 2024-11-09
**Good proposal**

**Rating:** 10
**Confidence:** 4

**Review:**

This is a very good proposal, well done.

Pros:
- There is a clear problem statement which motivates the need for your research.
- The proposed methodology is well structured and innovative. To use English as a pivot language is an interesting approach and you clearly outline how the translation process will work. How the dataset will be curated is also clearly outlined which strengthens your proposal.

Cons:
- It would have been nice if a more formal problem definition was included.
- It would also be nice if you described what baseline you would be comparing your model to. The metrics are very well described but is there another model you would compare your model to for those metrics?

Overall, very good and I look forward to seeing how your model performs.

---

### Official Review · ~Lu_Fan_DB1 · 2024-11-09
**Review of the Proposal: "Cross-Cultural Language Adaptation: Fine-Tuning Gemma 2 for Diverse Linguistic Contexts"**

**Rating:** 8
**Confidence:** 4

**Review:**

Summary:
This proposal focuses on improving translation quality between Chinese and Malay, especially in scientific texts, by fine-tuning Google’s Gemma-2 model with English as a pivot language. The approach leverages techniques like Supervised Fine-Tuning (SFT) with Low-Rank Adaptation (LoRA) and a multilingual neural machine translation (MNMT) model, aiming to enhance accessibility to scientific knowledge across low-resource languages.

Strengths:
Innovative Pivot-Based Methodology: The use of English as an intermediary language to improve low-resource language translations is a well-motivated approach, promising in scenarios where direct translations are challenging.
Domain-Specific Focus: Targeting biology-related texts enhances the model's real-world applicability, as scientific translation is often limited in low-resource languages.
Comprehensive Evaluation Plan: The proposal includes a well-rounded evaluation strategy using metrics like COMET, BLEU, and BERT-F1, which are suitable for assessing translation quality across multiple dimensions.
Areas for Improvement:
Dataset Expansion: Increasing the size and diversity of datasets, particularly with more real-world, domain-specific examples, would likely enhance the model’s performance and generalizability.
Potential Limitations: Briefly discussing challenges such as syntactic differences between Chinese and Malay or limitations in using English as a pivot could provide a clearer scope of the project.
Overall Assessment:
This proposal presents a thoughtful and structured approach to address cross-cultural language adaptation in low-resource settings. By refining the evaluation and addressing some practical challenges, the study could make a significant impact on multilingual accessibility in scientific literature.

---

### Official Review · ~Xiaoqian_Liu7 · 2024-11-10
**Clear Problem Statement and Methodology**

**Rating:** 10
**Confidence:** 3

**Review:**

The proposal "Cross-Cultural Language Adaptation: Fine-Tuning GEMMA 2 for Diverse Linguistic Contexts" presents a method to improve translation quality between Chinese and Malay, particularly for scientific papers, using English as a pivot language. The authors aim to leverage Google's GEMMA 2 model family to facilitate knowledge transfer from high-resource languages to low-resource ones, addressing the language bias on the web and promoting inclusivity in multilingual communication.

The methodology is sound, utilizing Supervised Fine-Tuning (SFT) with Low-Rank Adaptation (LoRA) for parameter-efficient model adaptation and developing a direct translation capability between the two languages through Multilingual Neural Machine Translation (MNMT). The authors also propose a curated dataset from scientific articles and synthetic data generation to focus on domain-specific vocabulary and sentence structures.

The evaluation plan includes a combination of human and automated metrics such as COMET, BLEU, BERT-F1, and CHRF, which are appropriate for assessing translation quality and domain-specific language preservation.

---

### Official Review · ~Michael_Hua_Wang1 · 2024-11-11

**Rating:** 8
**Confidence:** 4

**Review:**

The absence of a sufficient corpus of training data is a perennial problem when it comes to training ML models, and this absence directly contributes to poor performance of LLMs in languages other than those most commonly used online.

This project proposes to use the language with the most available data (English) as an intermediary to enhance translations between two other languages (in this case, Chinese and Malay). The mechanism by which this might be done is well explained, and it seems feasible to accomplish this.

However, as far as I can tell, the proposal does not directly address the issue of data paucity with respect to low-resource languages, as presumably the root model being used for translation would be subject to the same issues. More explicit discussion to link the project with the problem initially discussed is perhaps warranted.

---

### Official Review · ~Zhuofan_Sun1 · 2024-11-12
**Review of "Cross-Cultural Language Adaptation: Fine-Tuning Gemma 2 for Diverse Linguistic Contexts"**

**Rating:** 10
**Confidence:** 4

**Review:**

The work specifically targets low-resource languages in the context of scientific translations, using English as a pivot language.
Strengths:1.Clear Problem Definition: The proposal identifies a relevant problem: the lack of robust machine translation models for low-resource languages, particularly in scientific contexts. This addresses a gap in multilingual natural language processing, especially for languages like Malay, which often lack extensive datasets.; 2. Innovative Approach: The methodology is innovative in its use of English as a pivot language, allowing for incremental fine-tuning with LoRA and supervised fine-tuning (SFT). This modular approach allows adaptation without retraining the entire model, which is efficient and cost-effective. 3.Detailed Methodology: The authors present a well-defined, step-by-step methodology, starting from fine-tuning the model for Chinese-English translation and then moving on to English-Malay. The planned transition to a direct Chinese-Malay translation model without the English pivot demonstrates a clear progression in model development. 4.Domain-Specific Focus: The selection of biology-related scientific texts is a strategic choice. It allows the model to develop an understanding of specific vocabulary and structures, making it more effective for the intended application in scientific translation.
Overall, this proposal is well-structured and addresses a significant challenge in multilingual NLP. By focusing on scientific translation between low-resource languages, the authors contribute to making scientific knowledge more accessible across linguistic barriers. The methodology is well-thought-out, with appropriate technical sophistication and a strong foundation in prior work. Addressing the concerns around dataset size, pivot language limitations, and evaluation expansion would enhance the project’s impact and scalability.

---

### Official Review · ~Chumeng_Jiang1 · 2024-11-12
**Well-designed Pipeline**

**Rating:** 9
**Confidence:** 4

**Review:**

This proposal seeks to enhance multilingual machine translation between low-resource languages, currently focusing on Chinese and Malay, and with an emphasis on translating scientific texts. Leveraging Google’s GEMMA 2 model, the approach includes fine-tuning the model through an English pivot to achieve high-quality translations and eventually eliminate the need for an English intermediary. Key methodologies include supervised fine-tuning with LoRA for parameter efficiency and transfer learning for direct cross-linguistic adaptation. The project will use metrics like COMET and BLEU for evaluation.

 Strengths:
- **Methodological Clarity and Thoughtfulness:** The proposal is well-structured with a clear multi-step approach, encompassing pivot-based fine-tuning, transfer learning, and dataset curation. This detailed methodology reflects a deep understanding of translation model adaptation challenges.
- **Comprehensive Automated Evaluation Framework:** A strong set of evaluation metrics (COMET, BLEU, BERT-F1, and CHRF) has been proposed to ensure a comprehensive assessment of translation quality, considering both semantic similarity and technical language preservation.

 Weaknesses:
- **Unclear explanation of human evaluation process:** The combination of low-resource language and scientific texts is likely to make human evaluation quite challenging. I’m wondering how this part will be specifically designed.
- **Lack of Justification for English Pivot Removal:** Why would applying MNMT after the multi-step process lead to an improvement rather than a loss in performance? Theoretically, MNMT might perform worse than the proposed method.